# Extended pharmacodynamic responses observed upon PROTAC-mediated degradation of RIPK2

Alina Mares[1], Afjal H. Miah[1], Ian E.D. Smith[1], Mark Rackham[1], Aditya R. Thawani [1], Jenni Cryan[1], Pamela A. Haile[2], Bartholomew J. Votta[2], Allison M. Beal[2], Carol Capriotti[2], Michael A. Reilly[3], Don T. Fisher[4], Nico Zinn[5], Marcus Bantscheff [5], Thomas T. MacDonald[6], Anna Vossenkamper[6], Phoebe Dace[1], Ian Churcher[1], Andrew B. Benowitz [1], Gillian Watt [1], Jane Denyer[1], Paul Scott-Stevens[7] & John D. Harling [1✉]

Proteolysis-Targeting Chimeras (PROTACs) are heterobifunctional small-molecules that can promote the rapid and selective proteasome-mediated degradation of intracellular proteins through the recruitment of E3 ligase complexes to non-native protein substrates. The catalytic mechanism of action of PROTACs represents an exciting new modality in drug discovery that offers several potential advantages over traditional small-molecule inhibitors, including the potential to deliver pharmacodynamic (PD) efficacy which extends beyond the detectable pharmacokinetic (PK) presence of the PROTAC, driven by the synthesis rate of the protein. Herein we report the identification and development of PROTACs that selectively degrade Receptor-Interacting Serine/Threonine Protein Kinase 2 (RIPK2) and demonstrate in vivo degradation of endogenous RIPK2 in rats at low doses and extended PD that persists in the absence of detectable compound. This disconnect between PK and PD, when coupled with low nanomolar potency, offers the potential for low human doses and infrequent dosing regimens with PROTAC medicines.

[1] Medicine Design, GlaxoSmithKline, Medicines Research Centre, Gunnels Wood Road, Stevenage, Hertfordshire SG1 2NY, UK. [2] Innate Immunity Research Unit, GlaxoSmithKline, 1250 South Collegeville Road, Collegeville, PA 19426, USA. [3] Drug Metabolism and Pharmacokinetics, GlaxoSmithKline, 1250 South Collegeville Road, Collegeville, PA 19426, USA. [4] Drug Design and Selection, GlaxoSmithKline, 1250 South Collegeville Road, Collegeville, PA 19426, USA. [5] Cellzome, a GSK company, Meyerhofstrasse 1, 69117 Heidelberg, Germany. [6] Centre for Immunobiology, Blizard Institute, Barts and The London School of Medicine and Dentistry, Queen Mary University of London, E1 2AT London, UK. [7] Drug Metabolism and Pharmacokinetics, GlaxoSmithKline, Medicines Research Centre, Gunnels Wood Road, Stevenage, Hertfordshire SG1 2NY, UK. ✉email: john.d.harling@gsk.com

Targeted degradation of intracellular proteins using Proteolysis-Targeting Chimeras (PROTACs) represents a rapidly emerging technology with applications in both drug discovery and chemical biology[1–3]. PROTACs are hetero-bifunctional molecules designed to simultaneously bind to a target protein and to an E3 ligase complex to promote the formation of a ternary complex and the transfer of ubiquitin to surface lysine residues on the target protein, tagging it for degradation by the proteasome. Once the ubiquitin transfer event has occurred, a PROTAC molecule is available to form a new ternary complex, thus acting in a catalytic manner. This mechanism of action contrasts with traditional small-molecule inhibitors, where high levels of sustained occupancy are frequently required to elicit the desired pharmacology. Although PROTAC technology was first described nearly 20 years ago[4], there has been a dramatic increase in interest and activity catalysed by improvements in the embedded E3 ligase binders[5–8] which have delivered higher cell permeability. These improvements have increased cellular protein degradation potency of PROTACs from the high micromolar range to the low or even sub-nanomolar range[9–12].

While PROTAC technology represents an attractive approach to protein knock-down in vitro and in vivo as part of fundamental target biology investigations, its potential to deliver a new class of small-molecule degrader therapeutics could be transformative. The PROTAC event-driven catalytic mechanism of action of PROTACs can deliver potent cellular effects, and there may be additional advantages in cases where protein synthesis is slow. In these cases, PROTACs may enable a disconnect between pharmacokinetics (PK) and pharmacodynamics (PD) to provide extended duration of in vivo efficacy long after the PROTAC has been cleared from systemic circulation. This PK/PD disconnect may provide novel opportunities to develop medicines with low human doses and infrequent dosing regimens, although to date this potential has only been anticipated[13] with no rigorous preclinical in vivo data available to support this hypothesis to our knowledge.

Receptor-Interacting Serine/Threonine Protein Kinase 2 (RIPK2, also known as RIP2, RICK, CARDIAK, and CARD3) sits downstream of the pattern recognition receptors NOD1 and NOD2. NOD1 recognises the intracellular presence of the peptidoglycan fragment D-glutamyl-meso-diaminopimelic acid (iE-DAP) found predominantly in Gram-negative bacteria, whereas NOD2 recognises the muramyl dipeptide (MDP) component of peptidoglycans found in most bacteria[14,15]. Activation of this pathway results in transcriptional activation of multiple inflammatory cytokine genes, and its dysregulation is associated with several immune disorders such as inflammatory bowel disease (IBD)[16,17] as well as severe pulmonary sarcoidosis[18] and multiple sclerosis[19]. Protein turnover data for RIPK2 in primary immune cells generated via dynamic Stable Isotope Labeling with Amino Acids in Cell Culture (SILAC) labeling experiments indicates a half-life of generally ~50 h or longer[20,21] (Supplementary Fig. 1). Considering this prolonged half-life rate, we viewed RIPK2 as a suitable candidate protein to explore the potential for an extended PD response from PROTAC-mediated target degradation.

We report here a series of optimised PROTACs that promote the in vivo degradation of the kinase RIPK2 at low doses with RIPK2 degradation that persists beyond the detectable PK presence of the compound. Additionally, they inhibit inflammatory cytokine production in human primary cells and in inflamed colon tissue biopsies taken from ulcerative colitis (UC) and Crohn's disease (CD) patients. Further, we demonstrate that the RIPK2-degrading PROTACs act selectively at a proteomic level.

## Results

We have previously disclosed the potent and selective von Hippel-Lindau (VHL) based RIPK2 PROTAC **1** utilising an aminobenzothiazole-quinoline based RIPK2 binder[9]. To assess the effect of varying the E3 ligase binder, we prepared the analogous PROTACs **2** and **3** where the VHL binder was replaced by inhibitor of apoptosis (IAP) and cereblon-based E3 ligase recruiting moieties, respectively (Fig. 1).

The compounds were incubated with THP-1 cells for 18 h after which RIPK2 levels were determined using a capillary-based immunoassay ($n = 3$). All three PROTACs potently degraded RIPK2 in a concentration dependent manner (Fig. 1). The IAP-based PROTAC **2** was found to have a log half-maximal degradation concentration (pDC$_{50}$) value of $9.4 \pm 0.1$ and was more potent than either the equivalent VHL based PROTAC **1** (pDC$_{50}$ $8.7 \pm 0.1$) or the cereblon based PROTAC **3** (pDC$_{50}$ $8.6 \pm 0.4$). Despite the sub-nanomolar degradation potency of PROTAC **2**, it

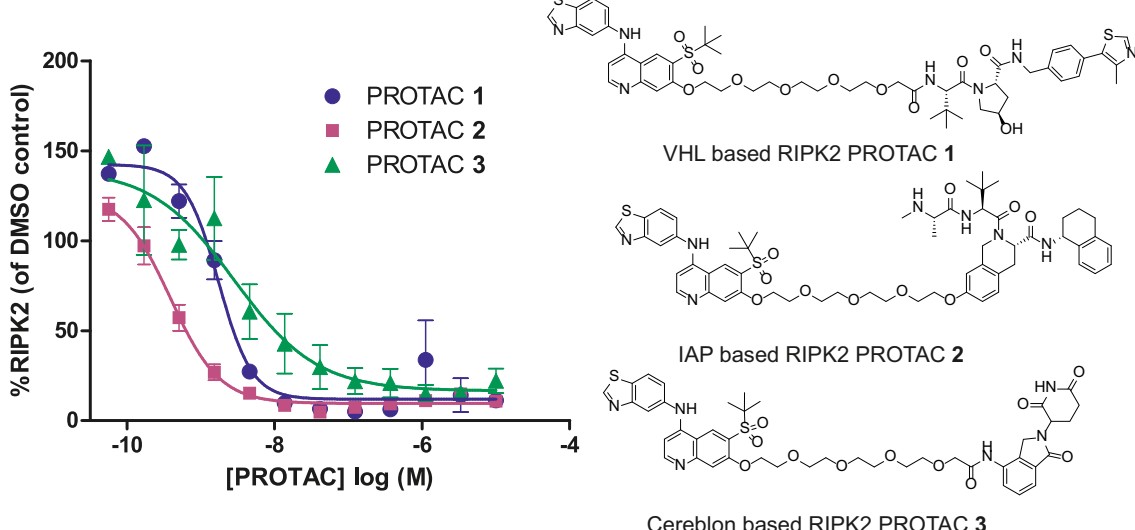

**Fig. 1 PROTAC molecules potently degrade RIPK2 in THP-1 cells.** We observed the degradation of RIPK2 in THP-1 cells relative to DMSO control upon treatment with analogous RIPK2 PROTACs employing VHL (**1**), IAP (**2**), and cereblon (**3**) E3 ligase recruiting moieties measured via a capillary-based immunoassay after 18 h treatment ($n = 3$).

has substantially weaker RIPK2 binding, with an $IC_{50}$ of 10 nM in a RIPK2 fluorescence polarisation binding assay. This discrepancy between the in vitro RIPK2 degradation and binding potency provides possible evidence for a catalytic mechanism of action of this compound.

While RIPK2 can be efficiently degraded by VHL, cereblon, and IAP based PROTACs and it is highly likely PROTACs based on all three ligases could be further optimized, we decided to focus on IAP based RIPK2 PROTACs for further studies. Consistent with PROTAC 2 being a potent degrader of RIPK2 in THP-1 cells, this compound also inhibited MDP-induced TNFα release in a human whole blood assay ($pIC_{50} = 7.95 \pm 0$). Nevertheless, the microsomal turnover of 2 in both rat and human microsomes was found to be very high (11 and 29 mL/min/g liver in rat and human microsomes, respectively), suggesting that this compound would have limited utility as an in vivo tool molecule. Additionally, this compound was found to have poor solubility as determined via a high-throughput precipitation assay using chemiluminescent nitrogen-specific detection (CLND)[22,23]. We hypothesised that reducing the lipophilicity of 2 would have a beneficial impact on reducing microsomal turnover and would also increase its solubility. The lipophilicity of test compounds was accessed via the $ChromLogD_{7.4}$ high-throughput chromatographic method, and PROTAC 2 was found to have a $ChromLogD_{7.4}$ of 6.1. The $ChromLogD_{7.4}$ technique affords a greater dynamic range than is feasible with the classical octanol-water method although a consequence of the scaling used in this technique is a positive offset of approximately two log units[24].

**Development of an initial in vivo RIPK2 PROTAC tool molecule.** The RIPK2-IAP PROTAC 4 (Fig. 2) identified in the course of our medicinal chemistry optimization program (manuscript in preparation) is significantly different from 2, possessing a different RIPK2 binder, IAP ligase binder, and linker. In human PBMCs, 4 degraded RIPK2 in a time dependent manner ($pDC_{50} = 7.9 \pm 0.2$, $D_{max}$ 69.2 ± 11.5% at 24 h, Supplementary Fig. 2) and inhibited L18-MDP (a synthetic derivative of MDP)-induced TNFα release ($pIC_{50}$ 8.0 ± 0.5 at 6 h). The inhibition of MDP-induced TNFα release in whole blood by 4 was the same in both rat and human ($pIC_{50}$ 7.7). A control PROTAC 5 (the enantiomer of PROTAC 4) maintains the RIPK2 binding potency (RIPK2 FP $pIC_{50}$ 6.7 ± 0.05) but in human PBMCs it does not degrade RIPK2 and showed more than 10-fold reduced activity in L18-MDP-induced TNFα release ($pIC_{50}$ 6.5 ± 0.5, Supplementary Fig. 2). The difference in potencies between PROTAC 4 and the control PROTAC 5 in the L18-MDP functional assay clearly illustrates the impact of the PROTAC mediated protein degradation.

Both the amino pyrazole-quinazoline RIPK2 binder and the IAP binder related to LCL-161 reduce the lipophilicity of PROTAC 4, and when incorporated with the linker shown, demonstrated low clearance in hepatocytes (<0.80 and <0.45 mL/min/g liver in rat and human liver hepatocytes, respectively).

Globally, these changes reduced the $ChromLogD_{7.4}$ of 4 to 4.2 and improved the high-throughput CLND solubility to >311 μM. The low turnover of PROTAC 4 rate in rat hepatocytes was found to translate into the in vivo setting, where the mean intravenous clearance in rats was less than a quarter of liver blood flow. This lower clearance coupled with a large volume of distribution resulted in an in vivo compound half-life of 6.9 h (Table 1).

The PK profile of compound 4 combined with its in vitro RIPK2-mediated functional inhibitory potency ($pIC_{50} = 7.7 \pm 0.05$) determined from the inhibition of L18-MDP-stimulated TNFα release in rat whole blood, suggested that this molecule would be suitable for the investigation of the PK/PD relationship resulting from in vivo RIPK2 degradation. Time course experiments were conducted in rats to determine the PK and associated PD profiles following a single subcutaneous dose; 20 mg/kg studied over 7 days and three dose levels of 1, 5, and 20 mg/kg studied over 14 days (Fig. 3).

High levels (>70%) of TNFα inhibition following ex vivo L18-MDP stimulation in rat blood were observed throughout the 7-day study following administration of a single dose of 20 mg/kg PROTAC 4, even though endogenous RIPK2 degradation measured by Western blot was only reduced by ~50% at 8 h post-dose, which then increased to >80% RIPK2 degradation when measured at both the 72 h and 168 h time-points highlighting the time-dependency of degradation (Fig. 3a). Although compound 4 contains a RIPK2 inhibitor, the in vitro L18-MDP data for the control PROTAC 5 would suggest that the high level of TNFα inhibition observed at the 8 h post-dose time point is mostly driven by the partial degradation of RIPK2 with only a small contribution from direct inhibition. At the later time-points, there was sustained RIPK2 degradation and inhibition of TNFα levels despite the compound levels declining to below 10 ng/mL, which is below the rat whole blood $IC_{90}$ value for inhibition of L18-MDP-stimulated TNFα release. These data indicate a disconnect between the PK and PD for this compound, with the efficacy at later timepoints being driven predominantly by RIPK2 degradation. As the initial study at 20 mg/kg was stopped after 168 h monitoring, a second study was also conducted with a longer monitoring period to establish the recovery of RIPK2 levels.

In the follow-up 14-day study (Fig. 3b, c), 40–60% degradation of RIPK2 was observed across all dose levels at the 8 h time point. The 20 mg/kg dose level data were consistent with the previous study. At 72 h post-dose, there was a clear dose-dependent

**Fig. 2 Structures for PROTACs 4 and 6 and control PROTACs 5 and 7.** We optimized PROTAC 2 to obtain PROTACs 4 and 6, as well as control compounds 5 and 7 which cannot recruit IAP.

**Table 1 Data for PROTACs 4 and 6 and control PROTACs 5 and 7 showing biochemical potency, $DC_{50}$, MDP challenge assay potency as well as rat intravenous PK parameters.**

| Cpnd | RIPK2 FP $pIC_{50} \pm$ SEM ($n$) | hPBMC $pDC_{50}$ /$D_{max} \pm$ SEM, 24 h ($n=3$) | hPBMC MDP $pIC_{50} \pm$ SEM, 6 h ($n=3$) | hWB MDP $pIC_{50} \pm$ SEM ($n$) | rWB MDP $pIC_{50} \pm$ SEM ($n$) | Rat Clearance mL/min/kg | Rat $Vd_{ss}$ L/kg | Rat $T\frac{1}{2}$ h |
|---|---|---|---|---|---|---|---|---|
| 4 | 6.6 ± 0.02 (8) | 7.9 ± 0.2/69.2 ± 11.5% | 8.0 ± 0.5 | 7.7 ± 0.05 (4) | 7.7 ± 0.16 (4) | 18 | 11 | 6.9 |
| 5 | 6.7 ± 0.05 (12) | Non-degrader | 6.5 ± 0.5 | – | – | – | – | – |
| 6 | 8.0 ± 0.05 (24) | 9.4 ± 0.2/94.3 ± 3.2% | 9.3 ± 0.03 | 8.5 ± 0.07 (14) | 8.5 ± 0.1 (4) | 10 | 7.6 | 16 |
| 7 | 8.4 ± 0.09 (4) | Non-degrader | 7.8 ± 0.2 | – | – | – | – | – |

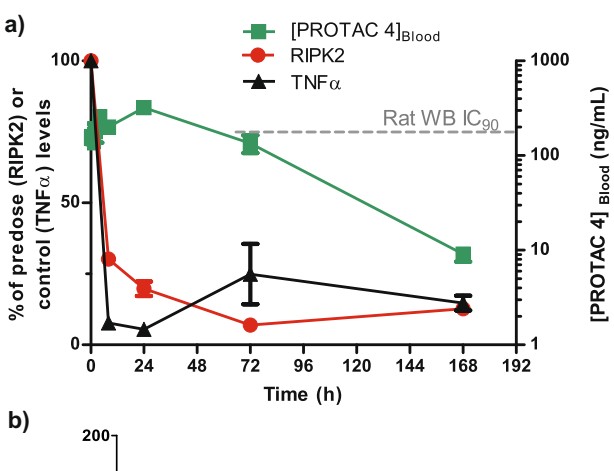

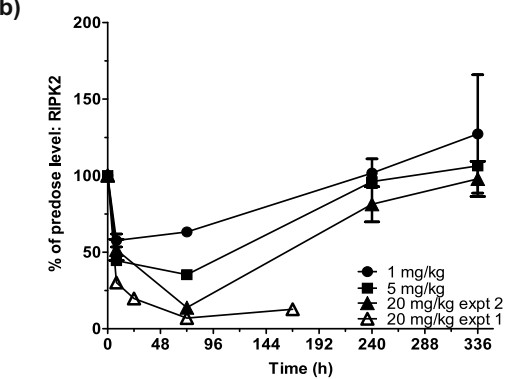

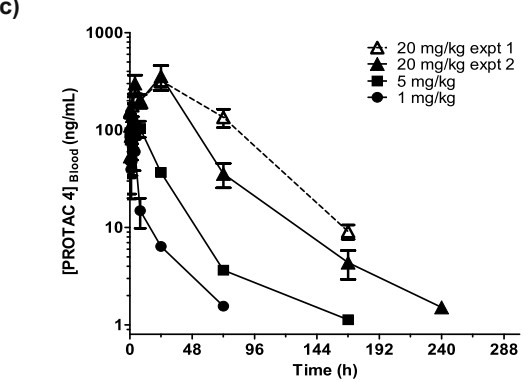

**Fig. 3 Pharmacokinetic and pharmacodynamic properties of PROTAC 4 in rats. a** Combined plot showing the relative time profiles for the pharmacokinetics and corresponding pharmacodynamics for PROTAC 4 after SC dosing at 20 mg/kg. **b** RIPK2 levels over time compared to individual animal pre-dose levels after SC dosing at 1, 5 and 20 mg/kg. **c** Rat exposure profiles for PROTAC 4. Data points are shown as mean ± s.e.m. ($n=5$).

degradation of RIPK2. The later time point of 240 h showed recovery of the protein levels which returned to baseline levels for each dose group. Overall, this data set suggests that the prolonged PD response is a function of both the PK profile of the compound and the slow synthesis rate of the target protein. The combined in vivo data set for compound 4 is consistent with RIPK2 possessing a half-life of approximately 2–3 days in rat.

**In vitro characterisation of the optimised RIPK2 PROTAC 6.** While 4 possesses improved physicochemical properties compared to 2 and a rat PK profile supportive of in vivo efficacy, the modest binding potency of 4 (RIPK2 $pIC_{50} = 6.6 \pm 0.02$) and cellular degradation potency ($pDC_{50} = 7.9 \pm 0.2$, $D_{max}$ 69.2 ± 11.5% at 24 h) impacts the dose required to deliver high levels of in vivo degradation of RIPK2. To address this limitation, further medicinal chemistry optimisation resulted in the identification of PROTAC 6 (Fig. 2), which contains an additional methylene spacer at the junction of the RIPK2 binding moiety and the linker, as well as a modified IAP binder. Crucially, the additional methylene group increased the RIPK2 inhibitory potency ($pIC_{50}$ 8.0 ± 0.05) more than 10-fold, which translated into significantly improved cellular degradation potency in human PBMCs (see Fig. 4) and human whole blood potency ($pIC_{50}$ 8.5 ± 0.07). Low turnover in hepatocytes was maintained in both rat and human (<0.8 and <0.45 ml/min/g liver, respectively), along with moderate lipophilicity and solubility (Chrom $LogD_{7.4}$ 3.6, CLND solubility 346 μM). In vivo rat PK data for 6 demonstrated reduced in vivo clearance compared to 4, which resulted in an extended half-life of 16 h for this compound (Table 1). Compound 6 also demonstrated a large window (>1000 fold) between RIPK2 degradation and the autoubiquitination-mediated degradation of cIAP1 in human PBMCs, and no effects on cell viability were detected except at concentrations >1 μM (Supplementary Fig. 3).

PROTAC 6 produced a concentration and time dependent decrease in RIPK2 protein levels in human PBMCs (Fig. 4a) such that the $pDC_{50}$ and maximal degradation ($D_{max}$) increased with longer incubation times, and a $pDC_{50}$ of 9.4 ± 0.2 and $D_{max}$ of 94.3 ± 3.2% was determined for 6 following 24 h incubation. PROTAC 6 completely inhibited TNFα, IL1β, IL-6, and IL-10 release following L18-MDP-stimulation with $pIC_{50}$ values >8.8, but only partially inhibited IL-8 release ($pIC_{50} = 8.5$ and $I_{max}$ ~ 50%; Fig. 4b). Since the in vitro activity of this compound is likely due to a combination of degradation and direct inhibition of the protein, the impact of RIPK2 degradation on the in vitro activity of 6 was assessed using an E3 ligase inactive control PROTAC 7 (the enantiomer of 6) which possesses the same RIPK2 binder but an inactive E3 ligase recruiting moiety. In contrast to 6, control PROTAC 7 did not significantly reduce RIPK2 protein levels (Fig. 4c) but was still found to inhibit TNFα release upon L18-MDP treatment with ~30-fold reduced potency ($pIC_{50} = 7.8 \pm 0.2$) and no significant difference in $I_{max}$ value (Fig. 4d).

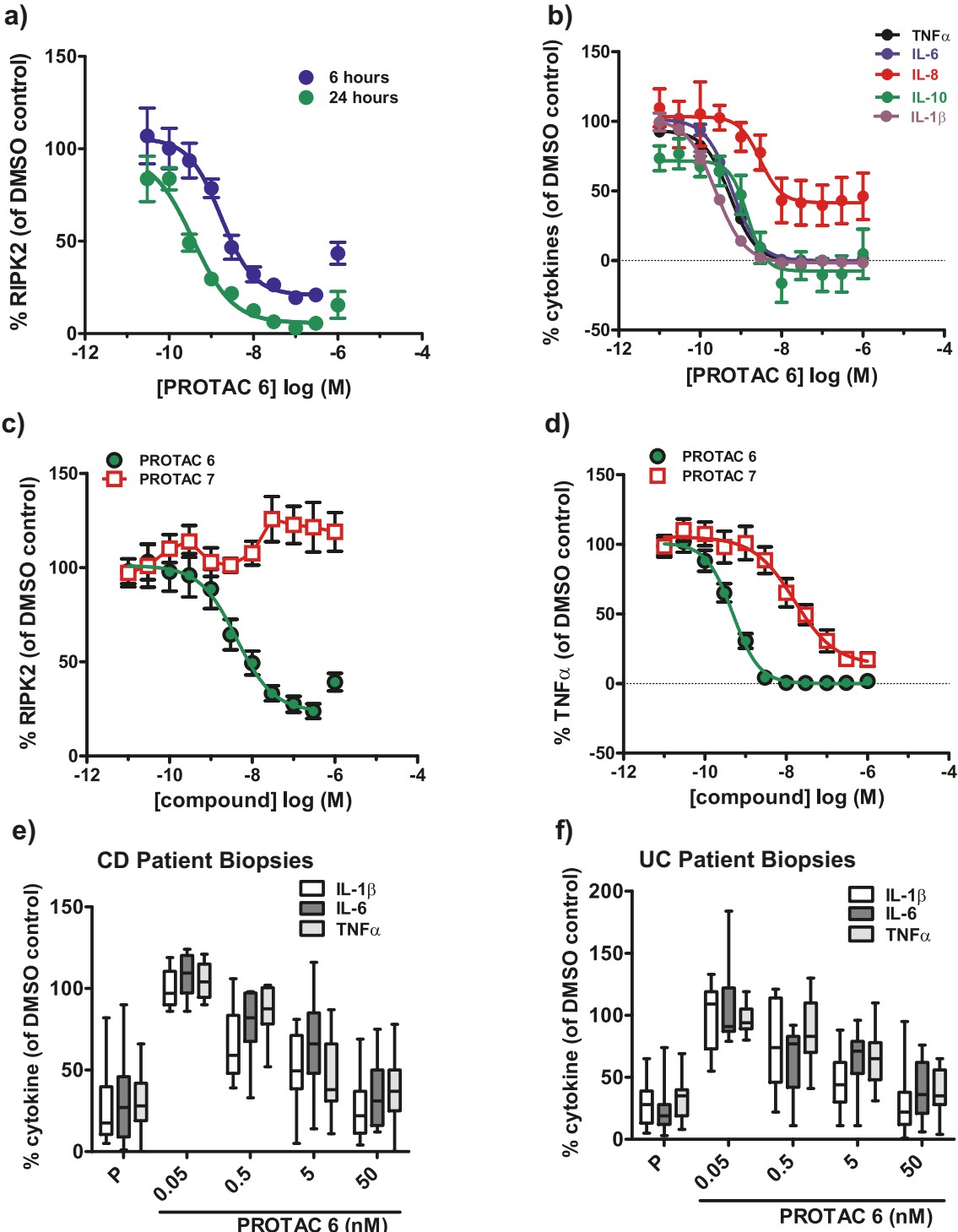

**Fig. 4 PROTAC 6 induces potent degradation of RIPK2 in human PBMCs and inhibits cytokine release in human disease tissue.** We tested the effect of PROTAC **6** on (**a**) RIPK2 levels at 6 and 24 h and (**b**) cytokine release following 3h pre-incubation and 3h L18-MDP stimulation in human PBMCs. Comparison of the effect of PROTAC **6** and negative control PROTAC **7** after 6h on (**c**) RIPK2 and (**d**) L18-MDP stimulated TNFα release in human PBMCs. Data are shown as mean ± s.e.m. ($n = 6$ except RIPK2 levels at 6h, $n = 5$). Spontaneous cytokine production from (**e**) CD and (**f**) UC disease patient intestinal biopsies ($n = 5$–19 per group) upon vehicle (DMSO) treatment (Ctrl) or treated with 1 µM prednisolone (P) or with increasing concentrations (0.05–50 nM) of PROTAC **6**.

Previous studies[25,26] have demonstrated that RIPK2 inhibitors can significantly inhibit spontaneous inflammatory cytokine release from inflamed intestinal tissue taken from UC and CD patients in an ex vivo culture system, supporting the hypothesis that RIPK2 kinase may be a key node in the signaling network connecting multiple pattern recognition receptors. Accordingly, PROTAC **6** was also tested in an ex vivo culture system using human biopsies taken from inflamed intestinal tissue of CD and UC patients (Fig. 4e, f and Supplementary Fig. 4). Upon treatment with **6**, a concentration dependent inhibition of

spontaneous IL-1β, IL-6, and TNFα release was observed, with a maximal effect obtained at 50 nM analogous to that achieved with 1 μM prednisolone. PROTAC **6** was equally effective at inhibiting spontaneous cytokine release across CD and UC patient biopsy samples with $IC_{50}$ values of ~1–3 nM. RIPK2 protein levels were also assessed by Western blot upon treatment of the CD and UC donor colon tissue with PROTAC **6**. This revealed a trend of concentration-dependent RIPK2 degradation compared to both control and prednisolone groups with a statistically significant reduction of RIPK2 ($p < 0.01$) observed in the UC biopsy group treated with 50 nM of PROTAC 6 although no significant reduction in RIPK2 levels was observed in the CD biopsy group. (Supplementary Fig. 4). However, due to the nature and time course of this assay system, it is not possible to distinguish the relative contributions of direct inhibition versus protein degradation to cytokine inhibition.

**PROTAC 6 shows in vivo efficacy beyond its detectable presence.** The potency of compound **6** in the rat whole blood functional L18-MDP challenge assay is $pIC_{50} = 8.5 \pm 0.1$. The PK/PD profile of this compound was investigated using a dose range of 0.005–0.5 mg/kg dosed SC in rats over 5 days. We observed that the single 0.5 mg/kg dose caused a significant reduction in RIPK2 protein levels, with $53 \pm 9\%$ RIPK2 degradation observed at 6 h post-dose, which increased to $78 \pm 5\%$ degradation at the 48 h time-point (Fig. 5a). This dose produced substantial inhibition (>70%) of TNFα levels upon ex vivo whole blood L18-MDP challenge, which was maintained over the duration of the study (Fig. 5b). The intermediate dose of 0.05 mg/kg did not have a statistically significant effect on measured levels of RIPK2 protein but significantly inhibited the release of TNFα upon ex vivo L18-MDP challenge by >60% over the first 48 h; at the later timepoints the TNFα levels were indistinguishable from control group. In contrast, the lowest dose of 0.005 mg/kg had no significant effect on RIPK2 or TNFα levels over the duration of the study. Blood concentrations of **6** (Supplementary Fig. 5) remained measurable throughout the study at the 0.5 mg/kg dose, for the first 24 h at the 0.05 mg/kg dose and for the first 6 h at the 0.005 mg/kg dose.

One of the potential advantages of PROTACs over inhibitors is the possibility for a cumulative efficacy effect with repeat compound administration in cases where the target protein is slowly synthesised in cells. To study this potential effect, once daily (QD) dosing compared to every 3 days (Q3D dosing were investigated with different dose levels (Fig. 5c–e). Compound **6** (QD dosing, 0.05 mg/kg/day) inhibited TNFα by ~60% at 6 and 24 h post-dose (measured immediately prior to the second dose) and progressed to >90% inhibition at the 54 h timepoint, which was 6 h after the third dose (Fig. 5c). The magnitude of RIPK2 degradation progressed over the study duration reaching >70% degradation at the last time point. At a higher dose of 0.5 mg/kg/day, profound inhibition (>90%) of TNFα release was obtained at the 24 h timepoint which was maintained following the next two doses (Supplementary Fig. 6). At this dose level, the observed degradation of RIPK2 increased from ~70% at 24 h to >90% at the last time point following the third dose. Analysis of terminal colon samples via Western blot at both doses revealed significant degradation of RIPK2 (Fig. 5f).

Using the 0.05 mg/kg dose level but dosing Q3D failed to show this cumulative PD effect (Fig. 5d). Repeating the Q3D dosing schedule at a higher dose level (0.15 mg/kg/day) produced maximal inhibition of TNFα at 72 h post dose, immediately prior to the second dose and 48 h after drug levels had dropped below the lower limit of quantification (LLQ). This response was maintained throughout the remainder of the study (Fig. 5e). This

was accompanied by a partial reduction in RIPK2 protein levels observed 24 h after the first dose, which was maintained with the subsequent doses.

In vivo L18-MDP administration to rodents results in an acute inflammatory response characterised by the rapid production of inflammatory cytokines[27]. Given the prolonged duration of PD response observed upon dosing PROTAC **6** in rats as characterized by inhibition of TNFα release following ex vivo L18-MDP challenge (Fig. 5), we sought to determine the PK profile and the kinetics of the PD response produced by a single dose of PROTAC **6** in blood and tissues (spleen and distal colon) in rats which received an intravenous dose of L18-MDP (150 μg/rat). The previously described PROTAC inactive control **7** was also included at the same dose level as **6** (0.3 mg/kg), since it possesses similar physicochemical properties and PK as **6**. Following L18-MDP administration, dosing of PROTAC **6** was found to inhibit the release of all tested inflammatory cytokines across compartments (Fig. 6). Serum cytokines were profoundly reduced (>75% inhibition) over the first three days and were significantly suppressed up to 6-10 days post-dose, while animals treated with the E3 ligase inactive control compound **7** were indistinguishable from the control animals except at the 72 h time-point, which was unexpected and inconsistent with the rest of the profile (Fig. 6a–c). Blood concentrations of **6** and **7** tracked closely and dropped below the LLQ after 24 h. In the spleen (Fig. 6d–f), the inhibition of TNFα and IL-8 was short-lived and returned to basal levels by the third day post-dose, whereas the release of these cytokines was significantly inhibited for up to six days post-dose in the distal colon (Fig. 6g–i).

**PROTAC 6 degrades RIPK2 with high selectivity.** To determine the degradation selectivity of **6**, we conducted studies employing multiple proteomics techniques across a range of cell types. Chemoproteomics profiling of **6** with kinobeads[28,29] using mixed protein extracts from HEK293, K-562, U-87 MG cells, and human placenta demonstrated high affinity of this compound for endogenous RIPK2 ($pK_d^{app} = 8.4$). Compound **6** was also found to be highly selective, with EPHA6 ($pK_d^{app} = 6.8$) and PAK4 ($pK_d^{app} = 6.4$) representing the only kinase off-targets with quantifiable affinity. No quantifiable affinity was measured for either RIPK1 or RIPK3 (see Supplementary data 2 file).

Compound **6** was further characterized by thermal proteome profiling[30,31] in U-87 MG and THP1 cells, measuring dose-dependent changes in thermal stability across the proteome. Consistent with the proposed mechanism of action of PROTAC **6**, temperature-independent reduction of RIPK2 levels was observed at concentrations ≥0.1 μM and *BIRC2* (cIAP1) levels were substantially reduced at concentration of **6** of >1 μM (Fig. 7a). This effect was found to be amplified upon prolonged incubation time (up to 240 min) with this compound. No other proteins were reproducibly affected in thermal stability or abundance by **6**, underpinning the exquisite selectivity of this compound.

Having characterized the cellular protein interactors of **6**, we also investigated the proteins degraded by this compound using the recently described multiplexed proteome dynamics profiling approach (mPDP)[32]. This technique can identify degraded targets through reduced levels in protein pools that were already present at the time dynamic SILAC labeling was initiated (mature proteins), as well as in the nascent proteins that were subsequently synthesised after SILAC labeling was conducted. U-87 MG or THP-1 cells were treated with PROTAC **6** at concentrations of 1 nM, 10 nM, 100 nM, and 1 μM, and were

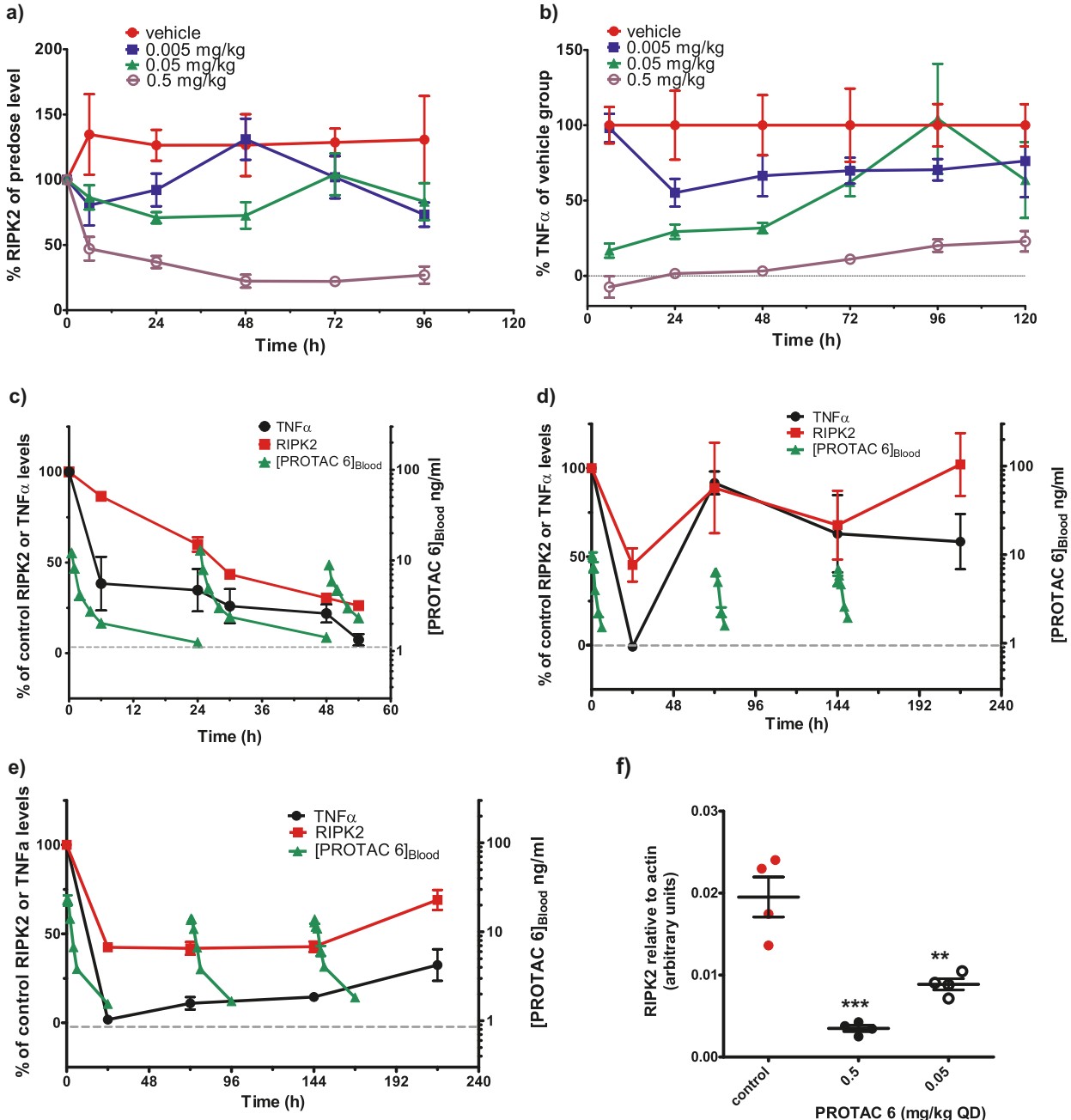

**Fig. 5 In vivo dosing of PROTAC 6 in rats caused the degradation of RIPK2 and a decrease in TNFα levels upon ex vivo L18-MDP challenge.** We determined the effect of dosing PROTAC **6** (0.005, 0.05 and 0.5 mg/kg SC) in rats on (**a**) RIPK2 levels compared to pre-dose levels and (**b**) TNFα levels following ex vivo L18-MDP challenge compared to the vehicle control group. Combined PK and PD time profiles are shown for 0.05 mg/kg (**c**) QD and (**d**) Q3D and (**e**) 0.15 mg/kg Q3D administered SC. **f** Effect of dosing PROTAC **6** (0.05 and 0.5 mg/kg QD SC) in rats on RIPK2 levels in colon compared to vehicle control group. Data are shown as mean±s.e.m and $n = 4$–5/group. P values calculated by ANOVA Dunett test comparing control vs. PROTAC (**P < 0.01, ***P < 0.001).

harvested after 6 and 24 h of treatment. A substantial reduction of mature RIPK2 protein levels was observed in U-87 MG cells after 6 h in the presence of 1 nM compound **6**, with no reduction observed in any other protein although *BIRC2* was not quantifiable (Fig. 7b). Selective degradation of RIPK2 was also observed with 10 nM compound **6** after 6 h; an apparent increase in PTMA (significantly regulated protein in Fig. 7b, c) at this time was not observed at higher concentrations or at the later 20 h timepoint. However, we observed a significant decrease in abundance of mature MAPK14 in addition to RIPK2 following

20 h incubation of both U-87 MG and THP-1 cells with compound **6** at concentrations ≥0.1 μM (see Supplementary Fig. 7). There were no changes in the abundance of PAK4, RIPK1 and RIPK3 in these experiments. The abundance of EPHA6 was not quantifiable. In addition to analysis of the protein extract, protein kinases were enriched using the kinobeads matrix[28]. This enabled investigation of changes in the abundance of kinases with low expression that would otherwise not be identified in the mass spectrometric analysis. The selective degradation of RIPK2 at both 6 and 20 h across both mature and nascent protein was

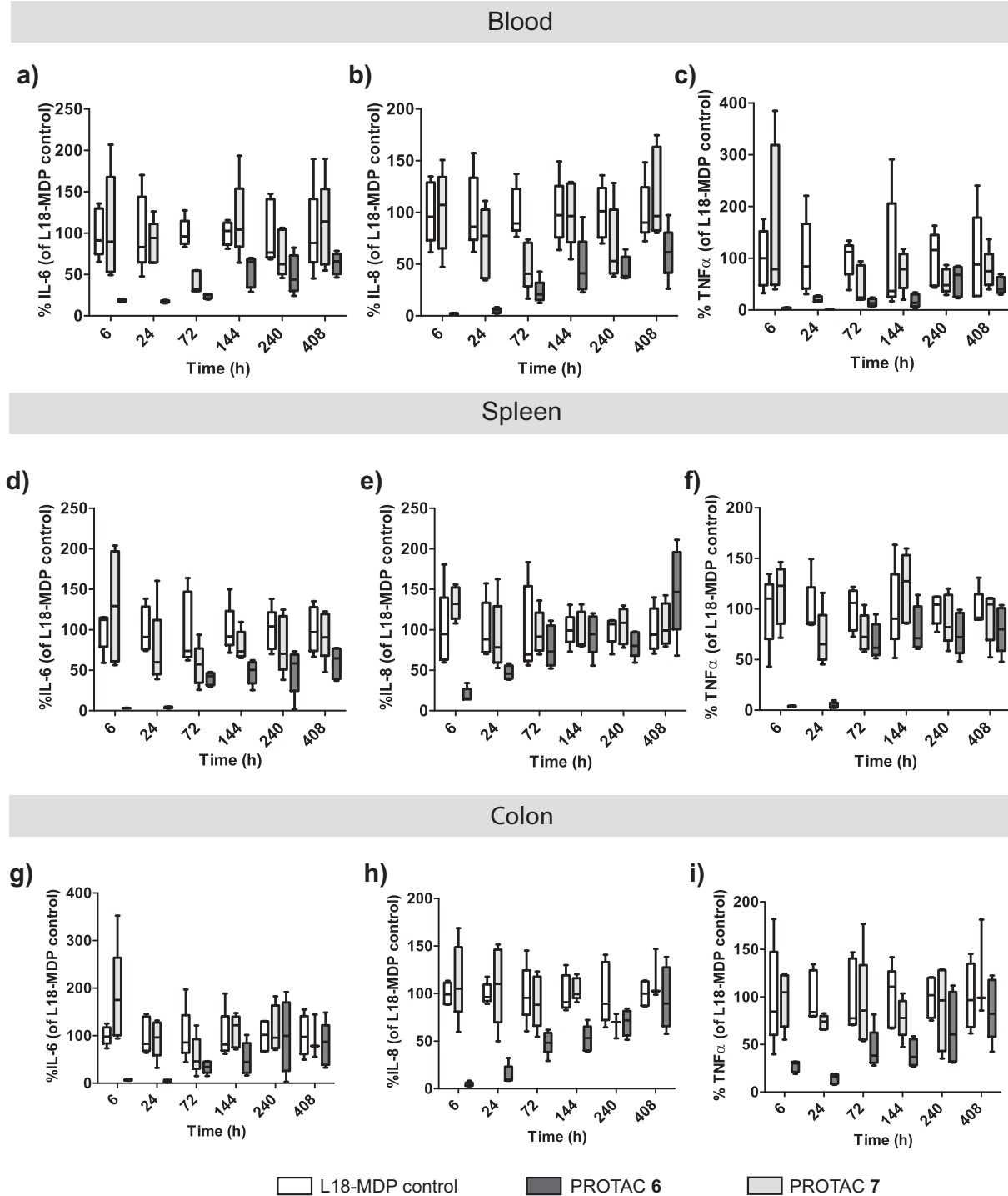

**Fig. 6 In vivo dosing of PROTAC 6 in rats suppressed cytokine release following intravenous L18-MDP challenge.** We determined the effect of dosing PROTAC **6** (0.3 mg/kg SC; black bars) or control PROTAC **7** (0.3 mg/kg SC; gray bars) in rats following intravenous L18-MDP challenge on the cytokine release in (**a–c**) blood, (**d–f**) spleen, and (**g–i**) colon. Data are plotted as mean ± s.e.m. ($n = 5$/group), with the timepoints shown indicating the length of PROTAC/vehicle treatment. This was followed by 2 h L18-MDP challenge at which point the cytokines levels were measured.

observed (Fig. 7d). However, this technique of kinase enrichment cannot distinguish between a PROTAC **6** mediated decrease in protein abundance and PROTAC **6** binding to the kinase at the ATP binding site but not inducing its degradation.

## Discussion
An appreciation of the significant opportunities afforded by recent developments in PROTAC-mediated targeted protein degradation

for both novel therapeutics and tools for chemical biology is clearly exemplified by the dramatic increase in publications over the last two years[1–3]. Many of these reports have presented in vitro studies that have demonstrated the importance of the ternary complex between the target protein, the PROTAC, and the E3 ligase complex in determining not only the degradation efficiency, but also introducing greater degradation selectivity as compared to the native binding selectivity of the target binding moiety of the

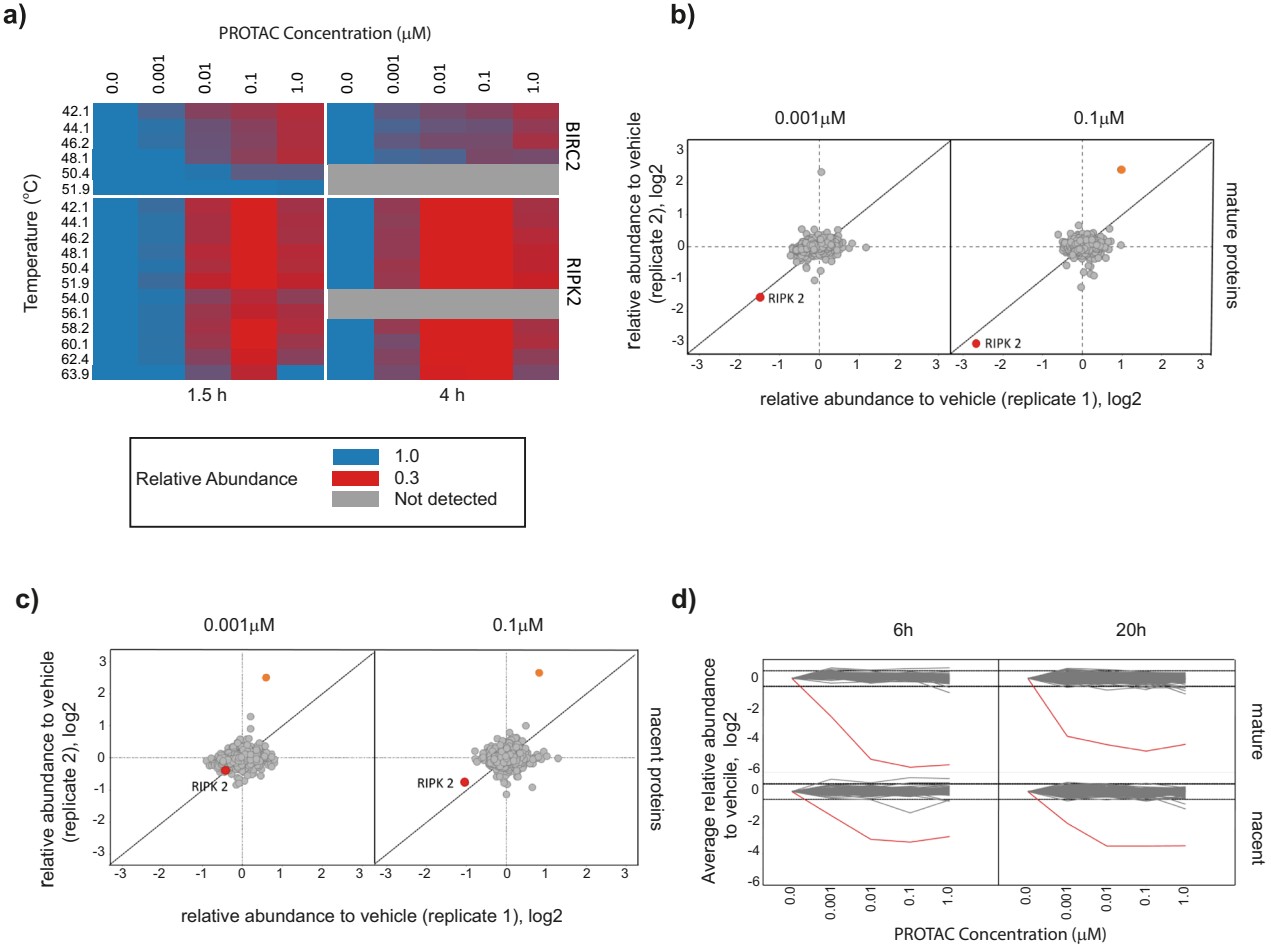

**Fig. 7 Thermal proteome profiling of PROTAC 6 demonstrates selective degradation of RIPK2 in cells. a** Thermal proteome profiling heatmap for RIPK2 and *BIRC2* (cIAP1) following treatment with PROTAC 6 across a concentration range of 0.001 μM to 1 μM and treatment durations of 0.5, 1.5, and 4 h (blue: 100% protein level, red: below 30%, gray: not identified) **b** mature proteins levels derived from mPDP profiling of PROTAC **6** at 0.001 μM and 0.01 μM in U-87 MG cells at 6 h, RIPK2 highlighted in red, orange indicates statistical significant regulation (PTMA) **c** nascent proteins levels derived from mPDP profiling of PROTAC **6** at 0.001μM and 0.01 μM in U-87 MG cells at 6 h, RIPK2 highlighted in red, orange indicates statistical significant regulation. **d** Line plot for protein dynamics profiling experiments separated for mature and nascent proteins following treatment of PROTAC **6** in U-87 MG cells followed by kinobeads enrichment of kinases in the cell extract; RIPK2 shown in red.

corresponding PROTAC[33–36]. In the current work, we began with a highly potent and selective RIPK2 inhibitors already available as a result of previous RIPK2 lead optimization efforts[25,37], and we were able to quickly demonstrate effective degradation of the target protein with PROTACs based on these inhibitors. It is noteworthy that the highly potent cellular degradation of RIPK2 by PROTAC **2** ($pDC_{50} = 9.4 \pm 0.1$) is achieved from a less potent biochemical RIPK2 $IC_{50}$ of 10 nM. This is indicative of the advantages that the catalytic, event-driven pharmacology that PROTACs can deliver when compared to traditional occupancy-based inhibitors.

The majority of protein degradation targets described to date are for oncology indications. As expected, much of the reported in vivo data for PROTACs that degrade these targets has been in mouse xenograft studies, and while significant tumor regression has been disclosed in several instances, these studies have generally not sought to develop a detailed PK/PD understanding[38]. A recent report has focused on the in vivo effects of PROTACs across preclinical species in various tissues, but the relationship of protein degradation to circulating drug concentrations is not discussed[39]. Since we were aware at the outset of this work that RIPK2 had a long-half life, particularly in primary cells, there was an expectation that we might observe enduring RIPK2 protein knock-down upon PROTAC-mediated RIPK2 in vivo degradation.

The first demonstration of this principle was observed with PROTAC **4** where a single SC administration of 20 mg/kg in rats produced sustained high levels of RIPK2 degradation and inhibition of ex vivo TNFα release for seven days post-dose despite drug levels decreasing below 10 ng/mL after 7 days, which is below the $IC_{90}$ for **4** in the L18-MDP challenge assay. Additionally, there was no discernible change in RIPK2 degradation or TNFα release inhibition between the 3-day and 7-day-timepoints despite the decline in circulating drug levels. This study clearly supports one of the anticipated potential advantages of PROTACs over small-molecule inhibitors by showing that infrequent dosing regimens are able to provide sustained efficacy in cases where the protein synthesis rate is slow. Studying the PK/PD response observed during a longer 14-day-study in rats allowed us to characterize the synthesis of RIPK2 protein back to basal levels following drug clearance, which was associated with a decrease in functional response as expected. This study allowed us to estimate the in vivo RIPK2 half-life in rat as 2–3 days.

PROTAC **6** is structurally similar to PROTAC **4**, although modifications to the linker afforded significantly increased binding potency for RIPK2 which translated into improved degradation potency and subsequent whole blood potency in both rat and human. The implication of this observation is that

the propylene-piperazine moiety may be considered as part of the RIPK2 binder with only the pyrazine ring constituting a short linker. Coupling the improved potency of PROTAC **6** with its long exposure in rat and the slow RIPK2 synthesis rate afforded efficacious in vivo protein degradation at very low doses. To our knowledge, the repeat dose PK/PD study also revealed a novel and highly attractive aspect of PROTAC pharmacology whereby repeated administration of sub-maximal efficacious doses caused a cumulative increase in protein degradation of RIPK2 without drug accumulation. Indeed, for the 0.05 mg/kg group dosed daily, PROTAC **6** $C_{max}$ concentrations never exceeded 15 ng/mL and were approximately at the LLQ (1 ng/mL) for most of the study. Further repeat dose studies with a less frequent dosing regimen provided further insights into the relationship of both dose and protein synthesis rates on maintaining reduced protein levels to elicit the PD response. The Q3D dosing of PROTAC **6** at 0.15 mg/kg provides some of the most compelling evidence for the extended PD response. Following the first dose, drug levels were maintained above the LLQ for 24 h. Nevertheless, at the 72 h timepoint, prior to the administration of the second dose, there was clear evidence of persisting RIPK2 degradation and inhibition of TNFα production in the absence of measurable drug levels

Overall, the extended in vivo PD responses in both blood and in tissues that we observed upon event driven PROTAC-mediated target degradation contrast significantly to the PD typically seen with traditional non-covalent inhibitors, where the pharmacological effect is often dependent on maintaining high levels of occupancy at trough concentrations. The high $C_{max}$ often associated with the high dose levels required to achieve the required exposure can be the cause of unwanted toxicities where selectivity windows are eroded[40]. In contrast, by reducing the need for high $C_{max}$ levels to achieve sufficient event driven therapeutic exposure, we postulate that therapeutic efficacy can be achieved with improved safety for PROTACs that degrade proteins with moderate to slow synthesis rates versus inhibitors.

Modern proteomic techniques can provide deep insights into compound selectivity in physiologically relevant cellular settings. For PROTAC **6** this has been achieved through a combination of kinobeads experiments, thermal proteome profiling and protein dynamics experiments. While size and molecular complexity have been implicated in binding promiscuity[41] it is clear from these proteomic experiments that **6** possesses an extremely selective binding profile with a very small number of off-target interactions. RIPK2 is the only detected degradation target for PROTAC **6** at concentrations below 0.1 μM.

Translation to the clinic represents the current frontier for the rapidly evolving PROTAC field. We have generated robust data through the PROTAC-mediated degradation of RIPK2 in rat in vivo and ex vivo studies, and also in human in vitro PBMCs and in human ex vivo inflamed biopsies, where we observe conclusive RIPK2 protein degradation and the inhibition of disease-relevant cytokine release. These results provide evidence that RIPK2 degradation may be a feasible approach to disease modification in CD and UC patients. Data from these biopsies also provides a useful foundation on which to build human dose predictions in conjunction with the in vivo efficacy data.

In summary, we have demonstrated how the event-driven, catalytic PROTAC mechanism of action coupled with slow protein re-synthesis rates affords opportunities to develop low dose medicines with infrequent dosing schedules and reduced exposure of drug, which may afford an improved safety window when compared to traditional inhibitors. We envision that this concept may become even more important in the future as additional PROTAC molecules are advanced into the clinic.

## Methods
Preparation and characterisation of compounds **1–7** is described in the Supplementary Information.

All animal studies were ethically reviewed and carried out in accordance with Animals (Scientific Procedures) Act 1986 (United Kingdom) or GSK Institutional Animal Care and Use Committee (United States of America) and the GSK Policy on the Care, Welfare and Treatment of Animals. The human biological samples were sourced ethically, and their research use was in accord with the terms of the informed consents under an IRB/EC approved protocol.

**Cell culture**. THP1 cells (ATTC, TIB-202) were cultured at a $3 \times 10^6$ cells/ml density in complete growth medium: RPMI 1640 medium containing GlutaMAX and 25 mM HEPES and supplemented with 10% fetal bovine serum, 100 U/mL penicillin and 100 μg/mL streptomycin.

**Antibodies**. Rabbit anti-RIPK2 (Cell Signaling, 4142), mouse anti-tubulin (Sigma, T8328), mouse anti-actin (Sigma, A2228), mouse anti-β-actin (Abcam, ab6276), rabbit anti-β-actin (Abcam, ab8227), mouse anti-vinculin (Abcam, ab129002), rabbit anti-cIAP1 (Cell Signaling, 7065), rabbit anti-cIAP2 (Cell Signaling, 3130), mouse anti-XIAP (BD Biosciences, 610717), donkey anti-rabbit IRdye 800CW (Licor, 926-32213), donkey anti-mouse IRdye 680RD (Licor, 926-68072), 20x anti-rabbit HRP conjugate antibody (Protein Simple, 043-426), anti-mouse HRP conjugate antibody (Protein Simple, 042-205), anti-rabbit HRP conjugate antibody (ProteinSimple, 042-206).

**Microsomal metabolic and hepatocyte stability**. Metabolic stability was performed at Cyprotex (UK). Pooled liver microsomes / cryopreserved pooled hepatocytes were purchased from a commercial supplier (BD Biosciences or Corning/InVitro Technologies).

*Microsomal $CL_{int}$*: Liver microsomes (final protein concentration 0.5 mg/mL), 0.05 M phosphate buffer pH 7.4 and test compound (final substrate concentration 0.5 μM; final DMSO concentration 0.25%) were pre-incubated at 37 °C prior to the addition of NADPH (final concentration 1 mM) to initiate the reaction. The final incubation volume was 500 μL. A minus cofactor control incubation was included for each compound tested where 0.05 M phosphate buffer pH 7.4 was added instead of NADPH (minus NADPH). Each compound was incubated for 0, 5, 15, 30, and 45 min. The control (minus NADPH) was incubated for 45 min only.

*Hepatocyte $CL_{int}$*: Williams E media supplemented with 2 mM L-glutamine and 25 mM HEPES and test compound (final substrate concentration 0.5 μM; final DMSO concentration 0.25%) were pre-incubated at 37 °C prior to the addition of a suspension of cryopreserved hepatocytes (final cell density $0.5 \times 10^6$ viable cells/mL in Williams E media supplemented with 2 mM L-glutamine and 25 mM HEPES) to initiate the reaction. The final incubation volume was 500 μL. An appropriate vehicle control was used. Each compound was incubated for 0, 5, 10, 20, 40, and 60 min. The control (vehicle) was incubated for 60 min.

All incubations were terminated by transferring 50 μL of incubate to 100 μL of acetonitrile containing internal standard at the appropriate time points. The termination plates were centrifuged at 2500 rpm for 20–30min at 4 °C to precipitate the protein. Following protein precipitation, samples were analysed using quantitative LC-MS/MS. A plot of ln peak area ratio (compound peak area/internal standard peak area) against time, yielded the gradient of the line. Subsequently, half-life ($t_{1/2}$) and intrinsic clearance ($CL_{int}$) were calculated using the equations below:

$$\text{Elimination rate constant (k)} = (-\text{gradient})$$
$$\text{Half-life } (t_{1/2})(\text{min}) = 0.693k$$
$$\text{Intrinsic clearance } (CL_{int})(\mu L/\text{min/mg protein}) = V \times 0.693 t_{1/2}$$

where $V$ = Incubation volume (μL)/Microsomal protein (mg).

Calculated intrinsic clearance values of control compounds were compared with the Cyprotex data library to confirm the appropriate activity of the microsomal/hepatocyte batch and data acceptance.

**In vivo studies**. *Rat intravenous PK studies*: PROTAC **4**; Intravenous PK studies were conducted in male Sprague Dawley (7–10 weeks old at start of dosing) rats, supplied by Charles River UK Ltd. Rats received a single intravenous (IV) bolus administration (2 mg/kg, 2 mL/kg, $n = 3$ rats) formulated as a solution in 5% DMSO:95% ($^v/_v$) 10% ($^w/_v$) Kleptose in saline. Serial blood samples were collected via jugular vein sampling into K₂EDTA tubes at various time points up to 24 h post-dose and diluted with an equal volume of water.

PROTAC **6**; Intravenous PK studies were conducted in male Han Wistar (7–9 weeks old at start of dosing) rats, supplied by Charles River laboratories, USA. Rats ($n = 3$) received a single intravenous (IV) bolus administration (1 mg/kg, 4 mL/kg, $n = 3$ rats) formulated as a solution in saline with the pH adjusted to 4.0. Serial blood samples were collected by femoral artery sampling into tubes containing an equal volume of water containing K₂EDTA at various time points up to 120 h post-dose.

Aliquots of diluted whole blood (50:50) were analysed using quantitative high-performance liquid chromatography with tandem mass spectrometric detection

(LC/MS/MS) following protein precipitation. All analytical runs met predefined run acceptance criteria. PK parameters were generated following non-compartmental analysis (Phoenix® WinNonlin® (Certara, L.P., St. Louis, MO)) of the resulting concentration-time data.

*Rat PK/PD studies*: On arrival, all rats were housed in a plastic solid bottom cage in groups of 2–3 with access to food and water ad libitum and under a 12 h light–dark cycle. Male CD or Wistar Han rats (7–9 weeks old, Charles River UK Ltd.) were randomly allocated to treatment groups upon arrival. Female CD rats (7–8 weeks old on arrival, Charles River Ltd.) were placed into groups using a randomization schedule based on body weight prior to the study start. All animals were acclimatized for a minimum of one week prior to the experiment commencing.

Subcutaneous PROTAC solution formulations were prepared on the day of dosing as outlined below and filtered through a 0.22 µm Millex PVDF filter (Millipore) prior to PROTAC administration. PROTAC **4** was dissolved in saline with the pH adjusted to 5.0 with sodium hydroxide (1 M). PROTAC **6** and **7** were dissolved in a vehicle of 25 mmol sodium acetate buffer in dextrose at pH 6.0 using diluted acetic or dissolved in saline with the pH adjusted to 5.0 with sodium hydroxide (1 M).

Blood samples were collected at selected time points post dose for pharmacokinetic (PROTAC drug levels) measurements were diluted 50:50 with purified water and 25 µl aliquots were analyzed using quantitative high-performance liquid chromatography with tandem mass spectrometric detection (LC/MS/MS) following protein precipitation. All analytical runs met predefined run acceptance criteria.

*Ex vivo stimulation of rat blood with L18-MDP for PD*: PPD blood samples taken prior to PROTAC administration or at fixed timepoints post PROTAC administration through the course of the study, were treated ex vivo with 100 ng/mL L18-MDP (Invivogen) or vehicle (0.1% DMSO) and incubated for 4 h. Following centrifugation, plasma was collected for cytokines analysis as detailed in cytokines assay section. Additionally, RIPK2 levels were determined in PBMCs isolated from blood and where collected, in sampled colon tissue as described in the text.

**In vivo L18-MDP acute inflammation model**. Following sub-cutaneous PROTAC **6** or **7** (0.3 mg/kg) or vehicle dosing, female rats received NOD2 pathway activation by treatment with L18-MDP at 150 µg/rat via an intravenous injection at the indicated timepoints. 2 h following L18-MDP challenge, rats were euthanized, and terminal blood samples and tissues collected for cytokine analysis. Cytokine levels were determined in serum generated from the terminal blood samples and in tissue lysates as detailed below.

**Isolation of rat peripheral blood mononuclear cells from blood**. Blood was mixed with optiPREP (Sigma) and following centrifugation at $1300 \times g$ the PBMCs layer was collected. For in vivo studies, RIPK2 levels were determined immediately via immunoblotting, whereas for in vitro studies PBMCs were treated as detailed in the in vitro treatment section.

**Isolation of human PBMCs from blood for in vitro experiments**. PBMCs were isolated from leukodepletion filter from platelet aphaeresis supplied by National Health System Blood and Transplant—NHSBT Apheresed at NHBST. The content of each leukodepletion filter was diluted with phosphate buffered saline—PBS (Gibco) and layered on top of HistoPaque 1077 (Sigma) and following centrifugation at 800 g the resultant PBMCs ring was collected.

**In vitro treatment of rat or human cells and blood**. Before treatment with compound or vehicle, THP1 cells, human and rat PBMCs were counted using Vi-Cell cell counter (Invitrogen) and adjusted to a final cell density of 3–3.3 cell/ml using complete growth medium. Cells or whole blood were treated with vehicle (0.1% DMSO) or fixed concentrations of compound up to 1 µM in half log step increments and incubated for the time period indicated in the figure.

Where the assessment of cytokines was performed, L18-MDP was added to the cell culture media (1 µg/ml) or to the rat blood samples (100 ng/ml) and samples were incubated for up to 4 h. For human blood, 250 ng/ml MDP (Invivogen) was added following compound treatment and the samples were further incubated for 5 h. Samples were them spun down at 1000 g and the cell culture supernatant, plasma or serum were collected for cytokines assessment, while cell pellets were used for RIPK2 and/or cIAP1, cIAP2, and XIAP levels analysis via immunoblotting.

**In vitro treatment of human tissue biopsy samples from CD and UC patients**. The human tissue samples were sourced ethically, and their research use was in accord with the terms of the informed consents under an IRB/EC approved protocol.

Colonic biopsies or tissues were obtained during endoscopy or via surgical resection from Crohn's Disease (CD) or Ulcerative Colitis (UC) patients (St Bartholomew's Hospital, London). All patients gave informed written consent. For surgical specimens, the mucosa was dissected away from the submucosa and cut into pieces of ~10–20 mg. Samples were incubated for 24 h in serum free HL1 medium

(Lonza) containing 2 mM L-glutamine, 100 U/mL penicillin, 100 µg/mL streptomycin and 10 µg/mL gentamicin and treated with either vehicle (0.1% DMSO), prednisolone (1 µM) as a positive control, or PROTAC **6** (0.05–50 nM in log unit increments). The supernatants were collected for cytokine measurement as detailed in cytokines assay section while tissues were used to determine RIPK2 levels.

**Determination of protein levels**. *Cells and tissue lysis*: Cell pellets were resuspended in CelLytic M cell lysis buffer (Sigma) containing protease inhibitor cocktail (Roche) while frozen rat tissues or UC and CD colon biopsies were ground using a ball mill and the resulted powder was resuspended in RIPA buffer (Thermo Scientific) with protease inhibitor cocktail. The samples were then spun down and the supernatants were collected to use for RIPK2 and or IAPs levels analysis.

*SDS-PAGE and immunoblotting*: Samples were mixed with the lithium dodecyl sulfate (LDS) sample buffer (Invitrogen) containing the reducing agent (Invitrogen) and separated onto a 4–12% polyacrylamide gel (Invitrogen). Transfer onto polyvinylidene difluoride (PVDF) Immobilon membranes (Millipore) was performed using the Trans-Blot wet transfer unit (Bio-Rad) containing Tris-glycine transfer buffer (25 mM Tris, pH 8.3 and 192 mM glycine) with 10% added methanol. Membranes were blocked against non-specific binding with Odyssey blocking buffer (Licor Bioscience), then incubated overnight with the anti-RIPK2 (1:500), followed by a 2 h incubation at room temperature with the antibody for the loading control (1:10,000). Following washing in PBS containing 0.1% Tween 20, membranes were incubated with anti-rabbit and anti-mouse IRdye secondary antibodies (1:5000 dilution) 1 h at room temperature. The infrared signal was detected using an Odyssey scanner CLX (Licor Biosciences) and densitometry was performed using the Image Studio Lite version 5.2 software (Licor Biosciences). The intensity of the infrared signal obtained with the RIPK2 antibody was quantified and normalized relative to the intensity of control protein signal.

*Simple Western immunoassay*: Cell lysates were mixed with 5× concentrated master mix (Protein Simple) and denatured at 95 °C on a heat block for 5 min together with the provided ladder (Protein Simple). Antibodies were diluted in the provided buffer as following: anti-RIPK2 (1:10 or 1:50), anti-cIAP1(1:10), anti-cIAP2 (1:10), anti-XIAP (1:10), mouse anti-β-actin (Abcam - 1:1000), rabbit anti-β-actin (Abcam - 1:50) and anti-vinculin (1:100). All reagents (blocking buffer, primary and secondary antibodies, streptavadin, luminol/peroxide, separation matrix, stacking matrix and water) were added to the Protein Simple plate containing the ladder and lysates and the plate was placed in a Protein Simple Western instrument and the Simple Western default size assay protocol was used. All samples were analyzed using the associated Compass software 3.1.7. The area under the peak values obtained for RIPK2, cIAP1, cIAP2 or XIAP were normalized relative to the control protein values.

*Cytokine measurement assay*: Cytokine levels were measured in rat plasma or rat PBMCs cell culture supernatants using rat V-Plex TNFα, in human plasma or human PBMCs cell culture supernatants with either TNFα tissue culture or 7-plex kit. Rat serum and tissues samples from the in vivo L18-MDP acute inflammation model were analyzed using the rat TNFα, Il-6 and IL-8 U-Plex kits. All kits provided by Meso Scale Discovery (MSD). Levels of cytokines spontaneously released from human colon biopsies cell culture supernatants were measured using Human IL-1 beta/IL-1F2 DuoSet ELISA (Bio-techne), Human IL-6 ELISA (Immunotools) and Human TNF-alpha DuoSet ELISA (Bio-techne). All samples were processed accordingly to the manufacturer's instructions. The MSD plates were analyzed using the MSD sector imager 6000 plate reader and cytokines concentration (pg/ml) was obtained by using the MSD DISCOVERY WORKBENCH software version 4.0.12.1

*Statistics and reproducibility*: All data were represented as percentage of the control sample: DMSO treated cells for the in vitro assays and the pre-dose or vehicle sample for the in vivo experiments.

In vitro experimental data were analyzed using a 4-parameter logistic equation in GraphPad Prism 5.0.4. to generate an estimate of the half-maximal concentration and maximal degradation, $DC_{50}$ and $D_{max}$ value respectively, for concentration-degradation relationships. Cytokine levels were analyzed in a similar manner to yield $IC_{50}$ and $I_{max}$ values. The mean sigmoidal relationship is shown superimposed on the data points displayed as mean ± s.e.m. Where analysis to a sigmoidal relationship was not possible, the data points are shown as mean ± s.e.m.

For in vivo studies, RIPK2:tubulin ratios and cytokine levels from each individual experiment were analysed either using a One-way ANOVA with Dunnet's post-test or a two way ANOVA using GraphPad Prism 5.0.4 to determine statistical significance with 95% confidence levels.

**Proteomics methods**. *Cell culture*: THP-1 cell cultures (ATCC TIB-202, male) were established in RPMI-based SILAC-L and SILAC-H medium. Cells were seeded in opposite SILAC label medium and incubated in presence of compounds or vehicle (DMSO) for the indicated time points at 37 °C, 5% $CO_2$. For harvesting, the cells were washed in PBS, pelleted, and snap-frozen in liquid $N_2$.

U87-MG cell cultures were established in in normal growth medium (DMEM, 10% dialyzed FBS, 5 ml NEAA, 5 ml sodium-pyruvate) or DMEM-based SILAC-L and SILAC-H medium. Cells were seeded in opposite SILAC label medium and incubated in presence of compounds or vehicle (DMSO) for the indicated time points at 37 °C, 5% $CO_2$. For harvesting, the cells were washed in PBS, pelleted, and snap-frozen in liquid $N_2$.

*Affinity enrichment chemoproteomics*: In mPDP experiments affinity enrichments were performed in lysates derived from PROTAC treated cells as described previously[28,29] by using kinobeads. Briefly, 100 µL (149 µg protein) cell extract were incubated with kinobeads (3.5 µL beads per sample) for 1 h at 4 °C. The beads were washed with lysis buffer and eluted with 20 µL 2× SDS sample buffer and subjected to SDS gel electrophoresis. Samples were further processed for LC-MS/MS analysis.

Affinity enrichment competition binding experiments were performed from amino-functionalized compounds coupled to NHS activated Sepharose (GE Healthcare) as described in H.C. Eberl et al. Chemical proteomics reveals target selectivity of clinical Jak inhibitors in human primary cells (submitted), briefly one milliliter of mixed lysate (HEK293/K-562/Placenta/U-87MG) (5 mg/mL protein concentration) was incubated with the respective PROTAC for 45 min at 4 °C over a range of concentrations followed by incubation with affinity matrix for 1 h. Beads were washed, and eluted with 50 µL of a 2× SDS sample buffer and subjected to SDS gel electrophoresis.

*2D-TPP experiments, in cells*: Experiments were performed as previously described[30]. DMSO (vehicle), PROTAC dissolved in DMSO at indicated concentrations were added to U87-MG cells (two plates for each concentration) and incubated for 0.5, 1.5 or 4 h as indicated at 37 °C and 5% $CO_2$. All following steps were performed as previously described[20]. The heat treatment was done at 12 temperatures spread across a temperature range of 42.0–63.9 °C. Experiments were analyzed as previously described[29,30] and (https://www.cell.com/cell/fulltext/S0092-8674(18)30174-0#secsectitle0070) with the exception that only proteins affected reproducibly at two independent treatment durations were reported.

*Mass spectrometry sample preparation*: Sample preparation was performed as described in (https://www.cell.com/cell/fulltext/S0092-8674(18)30174-0#secsectitle0070), Briefly, gel lanes were cut into three slices covering the entire separation range (~2 cm) and subjected to in-gel tryptic digestion[28]. TMT labeling was performed using the 10-plex TMT reagents, enabling relative quantification of 10 conditions in a single experiment. The labeling reaction was performed in 40 mM triethylammoniumbicarbonate, pH 8.53 at 22 °C and quenched with glycine. Labeled peptide extracts were combined to a single sample per experiment. Pre-fractionation was performed exactly as described by Savitski et al.[32].

*LC-MS analysis*: Analysis was performed essentially as described by Savitski et al.[32].

Briefly, samples were dried in vacuo and resuspended in 0.05% trifluoroacetic acid in water. Half of the sample was injected into an Ultimate3000 nanoRLSC (Dionex, Sunnyvale, CA) coupled to a Q Exactive (Thermo Fisher Scientific). Peptides were separated on custom-made 50 cm × 100 µm (ID) reversed-phase columns (Reprosil) at 55 °C. Gradient elution was performed from 2% acetonitrile to 40% acetonitrile in 0.1% formic acid and 3.5% DMSO over 2 h. Samples were online injected into Q-Exactive mass spectrometers operating with a data-dependent top 10 method. MS spectra were acquired by using 70,000 resolution and an ion target of 3E6. Higher energy collisional dissociation (HCD) scans were performed with 35% NCE at 35,000 resolution (at *m/z* 200), and the ion target settings was set to 2E5 so as to avoid coalescence[31].

**Peptide and protein identification**. Mascot 2.4 (Matrix Science, Boston, MA) was used for protein identification by using a 10 parts per million mass tolerance for peptide precursors and 20 mD (HCD) mass tolerance for fragment ions. The search database consisted of a customized version of the International Protein Index protein sequence database combined with a decoy version of this database created by using scripts supplied by MatrixScience. In isobaric mass tag based experiments carbamidomethylation of cysteine residues and TMT modification of lysine residues were set as fixed modifications. Methionine oxidation, and N-terminal acetylation of proteins and TMT modification of peptide N-termini were set as variable modifications. Multiplexed pulsed SILAC isobaric mass tags searches for light and heavy SILAC were performed independently. Carbamidomethylation of cysteine residues and TMT modification of lysine residues were set as fixed modifications for the SILAC light search. Carbamidomethylation, lysine $^{13}C_6$ $^{15}N_2$ with TMT, and arginine $^{13}C_6$ $^{15}N_4$ were set as fixed modifications in the heavy SILAC search. Methionine oxidation, N-terminal acetylation of proteins and TMT modification of peptide N-termini were set as variable modifications in both searches[32].

**Peptide and protein quantification**. Reporter ion intensities were read from raw data and multiplied with ion accumulation times (the unit is milliseconds) so as to yield a measure proportional to the number of ions; this measure is referred to as ion area[42]. Spectra matching to peptides were filtered according to the following criteria: mascot ion score > 15, signal-to-background of the precursor ion > 4, and signal-to-interference > 0.5. Fold changes were corrected for isotope purity as described and adjusted for interference caused by co-eluting nearly isobaric peaks as estimated by the signal-to-interference measure. Protein quantification was derived from individual spectra matching to distinct peptides by using a sum-based bootstrap algorithm; 95% confidence intervals were calculated for all protein fold changes that were quantified with more than three spectra.

**Reporting summary**. Further information on research design is available in the Nature Research Reporting Summary linked to this article.

## Data availability
The data underlying Figs. 1–6 is in the file Supplementary data 1. The raw data, including blot images are available from the authors on reasonable request. The source data for the proteomics plots in Fig. 7 are included in the files Supplementary data 2–4.

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

## Acknowledgements

We would like to thank the Royal London Hospital: Endoscopy department and Col-orectal Surgery department for the supply of inflamed tissue biopsies from UC and CD patients.

## Author contributions

A.H.M., I.E.D.S., M.R., A.R.T., P.A.H., and J.D.H. contributed to medicinal chemistry design and chemical synthesis. A.M., B.J.V., D.T.F., P.D., G.W., J.C. and J.D. contributed to in vitro biology experiments. T.T.M. and A.V. contributed human ex vivo biology. A.M., B.J.V., A.M.B., C.C., and G.W. contributed to in vivo biology experiments. M.A.R. and P.S-S. contributed to pharmacokinetic analysis. M.B. and N.Z. contributed the proteomic studies. J.D.H., A.B.B., and I.C. contributed to experimental design and program strategy.

## Competing interests

The authors declare no competing interests.
