## [Peer Review File · Communications Biology]

Reviewers' comments:

Reviewer #1 (Remarks to the Author):

Dear Editor

In this manuscript by Dr. Harling and colleagues 3 IAP based RIPK2 PROTACs and a 'cis' negative control compound are evaluated, in cellular, ex-vivo and in vivo experiments. Importantly, their most advanced compound, PROTAC 5, is shown to be an extremely potent and selective RIPK2 degrader in cellular experiments that is able to show prolonged pharmacodynamic (PD) effect in vivo at low doses. As their title suggests, they are able to successfully demonstrate sustained PD effect beyond the period of detectable compound exposure and thus demonstrate nicely an advantage that has been theoretically proposed for the emerging PROTAC modality.

There is currently a distinct lack of exemplified PK/PD data for PROTACs in the published literature and PD data generally (with or without the accompanying PK profiling) outside of the oncology field is a rarity. In that respect I would suggest that this body of work offers new and highly valuable findings for the targeted protein degradation and drug discovery fields.

There are however some areas that require rewriting and key additional data that needs to be provided to sufficiently support some of the statements and claims that are made. In particular I would see the following areas as particularly important:

1. Section beginning on line 112: I find this section adds little and the learnings are not clear. If the authors are claiming that based on this evidence IAP is a 'better' ligase to recruit for RIPK2 generally then I do not see the evidence shown as sufficient to support this claim. On the other hand, if they are claiming simply that this particular RIPK2 IAP based PROTAC was the best of the 3 PROTACs shown in this section I feel this adds little to their manuscript as they then switch to a different class of IAP-RIPK2 based PROTACs on which to base the main findings of the manuscript.

2. There are some statements made with respect to profiling of PROTAC 4 which lack sufficient evidence. Firstly, there is no cellular degradation data, most notably a timecourse and titration curve, which I feel is a fundamental starting point required to contextualise what is observed in vivo and particularly to help understand differences between PROTAC 4 and PROTAC 5 later in the manuscript. Furthermore, no negative control, such as a 'cis' compound that is a non-degrading form of PROTAC 4, is utilised. This approach is used to good effect in the following section comparing PROTAC 5 vs PROTAC 6, but is curiously absent here. The conclusions drawn on lines 144-147 cannot be made without comparison to a non-degrading control (i.e. a comparable inhibitor) and likewise the comments made on lines 165-168 would be at least better informed if a similar comparative study was provided. If it is not possible to provide in vivo data for a 'cis' control of PROTAC 4 then it would at least be important to show comparisons in cellular experiments.

3. The comments on lines 171 and 177 directly comparing binding affinity to RIPK2 with the ability to degrade RIPK2 better in vivo makes too many steps given that there is no cellular data to back up that binding affinity and degradation are correlated directly. Whilst an improvement in RIPK2 is unlikely to harm the chances of finding improvements in degradation, this is only one of a number of key factors required to deliver potent in vivo degradation. It could be more accurate to comment that differences in cellular DC50 are likely to impact the doses required in vivo, another reason why cellular data for PROTAC 4 would be an important addition. Basing this argument purely on binding affinity and inhibitory potency in the human whole blood assay seems a very inhibitor-centric approach and undermines many of the arguments made by the authors as to what differentiates degraders.

In addition, I would also ask the authors to address the following:

4. Line 42: Some reference(s) acknowledging those that worked towards identification of the IMIDs as CRBN ligands should be made here.
5. Line 44: more recent references would be good to include here, possible the very potent BET degraders recently published by Pillow et al, ChemMedChem DOI: 10.1002/cmdc.201900497
6. Line 89: Supplementary figure S2: Molecular weight markers on western blots for actin need to be shown clearer and labelled, there is also currently no marker shown near to RIPK2 either.
7. Line 111: It would be helpful to have a reference to support this statement regarding the pros and cons of ChromLogD.
8. Line 221: Please only state the statistically significant result observed in the UC group and state that the study in the CD donor group showed no effect. Also, whilst I assume it was availability of biopsy samples, it's not clear to understand why PROTAC 6 was not used as a comparison in this study to address the query postulated on line 224/225.
9. Line 367: The publication by Gadd et al in Nature Chem Bio in 2017 should be included here.

Reviewer #2 (Remarks to the Author):

The authors reported a series of optimized PROTACs for RIPK2 with nanomolar DC50 in vivo. A highly selective RIPK2 PROTAC was discovered and validated. They also proved extended PD responses in vivo beyond detectable PK presence, therefore demonstrated that the catalytic PROTAC mechanism can be utilized in therapeutics for a lower dose, lower frequency treatment compared to the traditional inhibitors.

I think Communications Biology is an odd choice of journal for this work. It is mostly about the PK/PD of their PROTACs, and might fit better in J. Med. Chem. or even Nature Medicine.

On the one hand this manuscript reads like an extremely thorough progress update. I begins with a PROTACs already reported (1), changes the E3 ligase ligand (2 and 3) then arrives at 5 after optimization. However, on the other hand, the work arrives at important conclusions regarding the catalytic properties of their compounds, and the durable responses that are possible when protein production in vivo is slow. I am aware of no other study that looks at PD PK parameters as thoroughly as this, and reaches a similarly profound conclusion.

I recommend accept with very minor changes.

Some minor points:

Figure 1: DC50 should have error bars (should be generated with S-curve simulation)

Authors should define all abbreviations (eg PRR, Q3D, CNLD)

Reviewer #3 (Remarks to the Author):

The authors developed a series of RIPK2 targeting PROTACs and examined their in vivo efficacy to show extended PD that persists even in the absence of detectable PRPTAC compound. The paper is clearly written, however, the following minor concerns should be addressed before its publication at Communicational Biology.

1. Figure 1A and S2, it will be nice for the authors to monitor the changes in expression levels of RIPK1 and RIPK3 as well to demonstrate the specificity of the RIPK2-targeting PROTACs.
2. Figure 4C, the authors used the inactive version of PROTAC6 as a control, it will be nice if the authors can explain the dominate negative effects of PROTAC6.
3. Figure 7B, the authors should clarify whether the expression of EPHA6 or PAK4 which has been shown to potentially interact with PROTAC5, can be affected by PROTAC5?

Please find below in red our responses to all the reviewers' comments and suggestions.

Reviewer #1

In this manuscript by Dr. Harling and colleagues 3 IAP based RIPK2 PROTACs and a 'cis' negative control compound are evaluated, in cellular, ex-vivo and in vivo experiments. Importantly, their most advanced compound, PROTAC 5, is shown to be an extremely potent and selective RIPK2 degrader in cellular experiments that is able to show prolonged pharmacodynamic (PD) effect in vivo at low doses. As their title suggests, they are able to successfully demonstrate sustained PD effect beyond the period of detectable compound exposure and thus demonstrate nicely an advantage that has been theoretically proposed for the emerging PROTAC modality.

There is currently a distinct lack of exemplified PK/PD data for PROTACs in the published literature and PD data generally (with or without the accompanying PK profiling) outside of the oncology field is a rarity. In that respect I would suggest that this body of work offers new and highly valuable findings for the targeted protein degradation and drug discovery fields.

There are however some areas that require rewriting and key additional data that needs to be provided to sufficiently support some of the statements and claims that are made. In particular I would see the following areas as particularly important:

1. Section beginning on line 112: I find this section adds little and the learnings are not clear. If the authors are claiming that based on this evidence IAP is a 'better' ligase to recruit for RIPK2 generally then I do not see the evidence shown as sufficient to support this claim. On the other hand, if they are claiming simply that this particular RIPK2 IAP based PROTAC was the best of the 3 PROTACs shown in this section I feel this adds little to their manuscript as they then switch to a different class of IAP-RIPK2 based PROTACs on which to base the main findings of the manuscript.

It was not our intention to claim that IAP is a superior E3 ligase to degrade RIPK2 and the text has been modified to make this clear. The purpose of the section beginning on line 112 was to briefly outline the optimisation strategy which will be discussed in detail in a focussed medicinal chemistry communication. Having said that, the reviewer's comments appear to relate to the section before line 112. The purpose of this section was to provide a link to our earlier paper and show that RIPK2 can be readily degraded by PROTACs incorporating multiple E3 ligases.

2. There are some statements made with respect to profiling of PROTAC 4 which lack sufficient evidence. Firstly, there is no cellular degradation data, most notably a timecourse and titration curve, which I feel is a fundamental starting point required to contextualise what is observed in vivo and particularly to help understand differences between PROTAC 4 and PROTAC 5 later in the manuscript. Furthermore, no negative control, such as a 'cis' compound that is a non-degrading form of PROTAC 4, is utilised. This approach is used to good effect in the following section comparing PROTAC 5 vs PROTAC 6, but is curiously absent here. The conclusions drawn on lines 144-147 cannot be made without comparison to a non-degrading control (i.e. a comparable inhibitor) and likewise the comments made on lines 165-168 would be at least better informed if a similar comparative study was provided. If it is not possible to provide in vivo data for a 'cis' control of PROTAC 4 then it would at least be important to show comparisons in cellular experiments.

We have now included *in vitro* data for a control of PROTAC 4. PROTAC 5 in the revised manuscript is the enantiomer of 4 which binds to RIPK2 with a similar *in vitro* potency (pIC_{50} 6.7) but does not degrade RIPK2. Additionally, we have demonstrated that the control PROTAC 5 possesses significantly reduced potency in the MDP-stimulated TNF α release assay in PBMCs (pIC_{50} 6.5) compared to the active PROTAC 4 (pIC_{50} 8.0). This demonstrates the significant benefits of the PROTAC mechanism of action. Given the reduced *in vitro* activity of PROTAC 5, which would likely reduce further in a whole blood setting we predicted that it would be very unlikely that we would be able to measure a response *in vivo* with this compound. These new data also allow the comments in lines 165-168 to be put into better context. We have also modified lines 144-147 to reflect the fact that direct inhibition will likely be contributing only a small fraction of the observed functional TNF α inhibition.

3. The comments on lines 171 and 177 directly comparing binding affinity to RIPK2 with the ability to degrade RIPK2 better *in vivo* makes too many steps given that there is no cellular data to back up that binding affinity and degradation are correlated directly. Whilst an improvement in RIPK2 is unlikely to harm the chances of finding improvements in degradation, this is only one of a number of key factors required to deliver potent *in vivo* degradation. It could be more accurate to comment that differences in cellular DC₅₀ are likely to impact the doses required *in vivo*, another reason why cellular data for PROTAC 4 would be an important addition. Basing this argument purely on binding affinity and inhibitory potency in the human whole blood assay seems a very inhibitor-centric approach and undermines many of the valid arguments made by the authors as to what differentiates degraders.

We agree that delivering potent *in vivo* degradation is multi-factorial. We also believe that it is important to focus on the functional consequence of the degradation as ultimately this is the most important endpoint. We have now included DC₅₀ data for PROTAC 4 which provides the link from RIPK2 binding potency to cellular DC₅₀ to functional inhibition of TNF α in PBMCs and whole blood as requested by the reviewer.

In addition, I would also ask the authors to address the following:

4. Line 42: Some reference(s) acknowledging those that worked towards identification of the IMiDs as CRBN ligands should be made here.

The work of Ito et al identifying CRBN as a primary target for thalidomide and the work of Fisher et al solving the structure of a DBB1-CRBN structure in complex with thalidomide have both been cited

5. Line 44: more recent references would be good to include here, possible the very potent BET degraders recently published by Pillow et al, ChemMedChem DOI: 10.1002/cmdc.201900497

Further references have been added

6. Line 89: Supplementary figure S2: Molecular weight markers on western blots for actin need to be shown clearer and labelled, there is also currently no marker shown near to RIPK2 either.

The n=1 data in S2 has been replaced by n=3 data generated using a capillary based immunoassay on a proteinsimple™ Sally Sue system.

7. Line 111: It would be helpful to have a reference to support this statement regarding the pros and cons of ChromLogD.

Reference 22 contains this information

8. Line 221: Please only state the statistically significant result observed in the UC group and state that the study in the CD donor group showed no effect. Also, whilst I assume it was availability of biopsy samples, it's not clear to understand why PROTAC 6 was not used as a comparison in this study to address the query postulated on line 224/225.

The lack of statistically significant reduction of RIPK2 levels in the CD group has been added to the manuscript. As postulated by the reviewer, availability of biopsy samples was limited which precluded the suggested comparison.

9. Line 367: The publication by Gadd et al in Nature Chem Bio in 2017 should be included here.

This reference has been added

Reviewer #2 (Remarks to the Author):

The authors reported a series of optimized PROTACs for RIPK2 with nanomolar DC50 in vivo. A highly selective RIPK2 PROTAC was discovered and validated. They also proved extended PD responses in vivo beyond detectable PK presence, therefore demonstrated that the catalytic PROTAC mechanism can be utilized in therapeutics for a lower dose, lower frequency treatment compared to the traditional inhibitors.

I think Communications Biology is an odd choice of journal for this work. It is mostly about the PK/PD of their PROTACs, and might fit better in J. Med. Chem. or even Nature Medicine.

On the one hand this manuscript reads like an extremely thorough progress update. I begins with a PROTACs already reported (1), changes the E3 ligase ligand (2 and 3) then arrives at 5 after optimization. However, on the other hand, the work arrives at important conclusions regarding the catalytic properties of their compounds, and the durable responses that are possible when protein production in vivo is slow. I am aware of no other study that looks at PD PK parameters as thoroughly as this, and reaches a similarly profound conclusion.

I recommend accept with very minor changes.

Some minor points:

Figure 1: DC50 should have error bars (should be generated with S-curve simulation)

The original Figure 1 was n=1 Western data. This has been replaced in the revised manuscript with a new experiment for Figure 1 with n=3 data using capillary-based immunoassay RIPK2 protein quantification using a protensimple® Sally Sue system.

Authors should define all abbreviations (eg PRR, Q3D, CNLD)

These have all now been defined

Reviewer #3 (Remarks to the Author):

The authors developed a series of RIPK2 targeting PROTACs and examined their in vivo efficacy to show extended PD that persists even in the absence of detectable PRPTAC compound. The paper is clearly written, however, the following minor concerns should be addressed before its publication at Communicational Biology.

1. Figure 1A and S2, it will be nice for the authors to monitor the changes in expression levels of RIPK1 and RIPK3 as well to demonstrate the specificity of the RIPK2-targeting PROTACs.

Kinobead experiments with PROTAC5 (PROTAC 6 in the revised manuscript) included in the manuscript demonstrate no quantifiable inhibition of either RIPK1 or RIPK3. Both RIPK1 or RIPK3 were detected in the proteome dynamics profiling studies with PROTAC 5. No changes in the abundance of either protein was observed under any of the conditions studied. We have now commented on this the revised manuscript.

2. Figure 4C, the authors used the inactive version of PROTAC6 as a control, it will be nice if the authors can explain the dominate negative effects of PROTAC6.

The apparent increase in RIPK2 levels at higher concentrations of control PROTAC 6 (PROTAC 7 in the revised manuscript) are not significant.

3. Figure 7B, the authors should clarify whether the expression of EPHA6 or PAK4 which has been shown to potentially interact with PROTAC5, can be affected by PROTAC5?

PAK4 levels did not change in proteome dynamic profiling studies with PROTAC 5 (PROTAC 6 in the revised manuscript). EPHA6 was not detected in these studies. These observations have been added to the revised manuscript.

REVIEWERS' COMMENTS:

Reviewer #1 (Remarks to the Author):

Many thanks for addressing all of my points. I enjoyed reading this manuscript and would be happy to recommend it's publication.

Reviewer #3 (Remarks to the Author):

The authors have addressed most of the raised concerns during this round of revision.